# Slt2-MAPK/RNS1 Controls Conidiation via Direct Regulation of the Central Regulatory Pathway in the Fungus *Metarhizium robertsii*

**DOI:** 10.3390/jof8010026

**Published:** 2021-12-28

**Authors:** Yamin Meng, Xingyuan Tang, Yuting Bao, Mingxiang Zhang, Dan Tang, Xing Zhang, Xiaoxuan Chen, Weiguo Fang

**Affiliations:** MOE Key Laboratory of Biosystems Homeostasis & Protection, Institute of Microbiology, College of Life Science, Zhejiang University, Hangzhou 310058, China; 11807001@zju.edu.cn (Y.M.); txy627@126.com (X.T.); baoyuting@zju.edu.cn.com (Y.B.); 2015113053@stumail.nwu.edu.cn (M.Z.); 21807030@zju.edu.cn (D.T.); zhangxing_hubu@163.com (X.Z.); xuanzi@zju.edu.cn (X.C.)

**Keywords:** *Metarhizium*, conidiation, MAPK, *BrlA*, phosphorylation, RNS1, fungi

## Abstract

Ascomycete fungi usually produce small hydrophobic asexual conidia that are easily dispersed and essential for long-term survival under a variety of environmental conditions. Several upstream signaling regulators have been documented to control conidiation via regulation of the central regulatory pathway that contains the transcription factors BrlA, AbaA and WetA. Here, we showed that the Slt2-MAPK signaling pathway and the transcription factor RNS1 constitute a novel upstream signaling cascade that activates the central regulatory pathway for conidiation in the Ascomycetes fungus *M. robertsii*. The *BrlA* gene has two overlapping transcripts *BrlAα* and BrlAβ; they have the same major ORF, but the 5’ UTR of *BrlAβ* is 835 bp longer than the one of *BrlAα*. During conidiation, Slt2 phosphorylates the serine residue at the position 306 in RNS1, which self-induces. RNS1 binds to the *BM2* motif in the promoter of the *BrlA* gene and induces the expression of the transcript *BlrAα*, which in turn activates the expression of the genes *AbaA* and *WetA*. In conclusion, the Slt2/RNS1 cascade represents a novel upstream signaling pathway that initiates conidiation via direct activation of the central regulatory pathway. This work provides significant mechanistic insights into the production of asexual conidia in an Ascomycete fungus.

## 1. Introduction

Ascomycete fungi usually produce small hydrophobic asexual conidia that are easily dispersed in the environment [1,2]. During conidiation, fungal hyphae differentiate into specific structures called conidiophores, which is an important species-specific feature and can be used for fungal classification [3,4]. Although, Ascomycete fungi have a variety of shapes of conidiophores, the mechanisms for regulating conidiation seems to be rather conserved; three central regulators, the transcription factors BrlA, AbaA and WetA, have been documented in many fungi to regulate conidiation [5]. BrlA is the primary regulator and is essential for the initiation of conidiophore development [6]. The *abaA* expression is dependent on BrlA, and it regulates expression of genes associated with late phase of conidiation such as the differentiation of phialides [6]. The *wetA* gene is activated by AbaA in the middle to late phases of conidiation and is essential for the completion of conidiation [5,7]. BrlA, AbaA and WetA have thus been proposed to constitute a central regulatory pathway that coordinates expression of conidiation-specific genes for conidiophore formation and conidial maturation [6].

It has been documented in some *Aspergillus* fungi that *BrlA* has two overlapping transcripts, *BrlAα* and *BrlAβ*, and the former begins within the intron of *BrlAβ* [8]. The ORF in *BrlAβ* encodes the same polypeptide as *BrlAα* except that BrlAβ includes an additional 23 amino acids at the NH terminus, and BrlAα and BrlAβ appear to have redundant functions [8]. The *BrlAβ* transcript is transcribed in vegetative cells prior to induction of conidiation, and the translation of a uORF in the 5′ untranslational region (5′ UTR) represses translation of the major ORF [9]. The initiation of conidiophore formation is achieved by the expression of *BrlAα* and the removal of the translational block imposed by the uORF in *BrlAβ* [7,9]. The expression of *BrlA* responds to multiple regulatory inputs and is thus subjected to complex regulation. In *Aspergillus* fungi, the expression of *BrlA* is controlled by various upstream regulators, including G protein signaling pathway, Ras signaling pathway, Calcineurin signaling pathway, FlG-mediated conidiation pathway and other components such as LaeA and Velvet complex, and transcription factors VosA and NsdD [3,10,11,12,13]. In *Aspergillus nidulans*, AbaA was found to negatively repress the expression of *BrlAβ*, but no solid evidence has been reported to support that AbaA directly regulates *BrlAβ* transcription [7,14]. Therefore, how these upstream signaling pathways activate the expression of *BrlAα* and *BrlAβ* remains to be determined. In other Ascomycete fungi, such as *Beauveria bassiana* and *Trichoderma* fungi, several upstream components such as the velvet family proteins and VosA have also been documented to control the central regulatory pathway [15,16]; yet again, how they regulate BrlA expression remains to be explored.

The Ascomycete fungus *Metarhizium robertsii* is an insect pathogen and a beneficial plant symbiont and has thus been developed as an environmentally friendly mycoinsecticides and biofertilizers [17,18,19]. *M. robertsii* has been used as a model to study the pathogenicity and development of insect pathogenic fungi [20]. Conidium is an active constituent of fungal insecticides and biofertilizers. An understanding of the regulatory processes involved in conidiation is essential to the commercial development and improvement of these fungal agents [17]. The regulation of conidiation in the Sordariomycetes fungus *M. robertsii* appears to be conserved to the phylogenetic distant Eurotiomycetes fungi including the *Aspergillus* spp. In *M. robertsii*, the transcription factors BlrA, AbaA and WetA also constitute a central regulatory pathway for controlling conidiation [21]. The upstream signaling component FlbA is also essential for conidiation [22]. As in other fungi, the mechanisms underlying the control of the central regulatory pathway during conidiation in *M. robertsii* remain to be fully understood. In a previous study, we found that the Slt2-MAPK cascade regulated conidiation by this fungus [23]. In this study, we further showed that Slt2-MAPK and the transcription factor RNS1 constitute an upstream cascade that controls the central regulatory pathway via direct induction of the transcription of *BrlAα*.

## 2. Material and Methods

### 2.1. Fungal and Bacteria Strains

The *M. robertsii* strain ARSEF 2575 was obtained from the Agricultural Research Service Collection of Entomopathogenic Fungi (Department of Agriculture, Ithaca, NY, USA). *Escherichia coli* DH5α and *Agrobacterium tumefaciens* AGL1 were used for plasmid construction and fungal transformation, respectively, as previously described [24].

### 2.2. Assays of Colony Growth and Conidiation

Assays of colony growth, conidiophore formation and conidial yield on PDA (potato dextrose agar) plates were assayed as previously described [21]. Briefly, 5 μL of conidial solution (1 × 10^7^ conidia/mL) was inoculated on the center of a PDA plate (diameter = 90 mm), which was then incubated at 26 °C for 14 d. The colony diameter was measured daily. For conidiophore observation and conidial yield determination, 100 μL of the conidial solution (1 × 10^7^ conidia/mL) was evenly spread onto a PDA plate, which was then incubated at 26 °C. At three days post-inoculation, conidiophores were microscopically observed, and at 15 days, the conidial yield was measured.

### 2.3. Assays of Tolerance to Abiotic Stresses

Assays of conidial tolerance to UV radiation, oxidative stress, hyperosmotic stress and cell wall-disturbing agent were conducted as previously described [23,25]. To assay the tolerance to UV radiation, conidia were inoculated into a Petri dish (diameter = 30 mm) containing 3 mL of the medium 1/2 SDY (Sabouraud dextrose broth supplemented with 1% yeast), and were then exposed to a 312 nm (280–320 nm) UV-B wavelength at 0.2 J cm^−2^ in a Bio-Sun++ chamber (VilberLourmat, Marne-la-Vallée, France) followed by incubation at 26 °C. The conidial germination was recorded every 2 h using an inverted microscope (Leica, Wetzlar, Hesse-Darmstadt, Germany). Tolerance to hyperosmotic, oxidative stress or a cell wall-disturbing agent was assayed by measuring conidial germination in the 1/2 SDY medium supplemented with 0.75 m of KCl, 3 mm of H_2_O_2_ and 1 mm of Congo Red (Sigma–Aldrich, St. Louis, MO, USA), respectively. The germination was checked every 2 h. The regular 1/2 SDY medium was used as a control (unstressed). The relative germination inhibition caused by a given stressor on a strain was calculated as (Gc-Gt)/Gc, where Gc and Gt represent the GT_50_ (time (hours) required for 50% conidia to germinate) of the stressed and unstressed conidia (the control), respectively. All assays were repeated three times with three replicates per repeat.

### 2.4. Yeast Two-Hybrid Assay

Yeast two-hybrid assay was conducted as previously described [26]. The coding sequences of RNS1 and Slt2 were cloned by PCR and inserted into the *Eco*R I/*Bam*H I sites of the plasmid pGADT7 and pGBKT7, respectively, to produce plasmid pGADT7-RNS1 and pGBKT7-Slt2. The primers used in the study were summarized in Appendix A. The plasmid pGADT7-RNS1 was transformed into the Y187 cell, and pGBKT7-Slt2 into the Y_2_HGold cell. The yeast competent cells were prepared using the Yeast-maker Yeast Transformation System 2 Kit (Takara Bio, Tokyo, Japan). The resulting strains from mating were grown on the medium (SD-His-Ade-Leu-Trp) supplemented with X-α-gal and AbA (Aureobasidin A (Takara Bio, Tokyo, Japan)) to assay the physical interaction between Slt2 and RNS1. The autoactivation of Slt2 was analyzed by growing the Y_2_HGold cells containing pGBKT7-Slt2 on the medium (SD-/His/-Ade/-Trp) with X-α-gal. The yeast two-hybrid and autoactivation assays were repeated three times.

### 2.5. Co-IP Assays

Coimmunoprecipitation (Co-IP) assay was conducted as previously described [27]. To assay the in vivo interaction between the RNS1 and Slt2, the coding sequence of Slt2 was first amplified by PCR and inserted into the master plasmid pPK2-bar-Ptef-MYC [27] to produce the plasmid pPK2-bar-Ptef-Slt2-MYC, which was then transformed into the strain *WT-RNS1-FLAG* [28] to result in the strain *RNS1-FLAG/Slt2-MYC* that simultaneously expressed the fusion proteins Slt2::Myc and RNS1::FLAG. As a control, the plasmid pPK2-bar-Ptef-MYC was also transformed into the strain *RNS1-FLAG* to produce the strain *RNS1-FLAG/MYC.* Fungal protein extraction and immunoprecipitation analysis were conducted as previously described [28].

### 2.6. Phos-Tag Assays

Protein extraction and Phos-tag analysis were conducted as previously described [28]. To assay RNS1 phosphorylation by Slt2, in addition to the strain *WT-RNS1-FLAG*, the plasmid pPK2-sur-Ptef-RNS1-FLAG was also transformed into the deletion mutant *∆**Slt2* to produce the strain *∆**Slt2-RNS1-FLAG.* We also constructed a mutant of the fusion protein RNS1 (RNS1^S306A^) by changing the serine reside at position 306 in the RNS1 protein to alanine using the site mutagenesis kit (NEB, Ipswich, MA, USA). The coding sequence of RNS1^S306A^ was inserted into plasmid pPK2-sur-Ptef-FLAG to produce pPK2-sur-Ptef- RNS1^S306A^ –FLAG, which was then transformed into WT and *∆**Slt2* to produce the strains *WT-RNS1^S306A^-FLAG* and *∆Slt2-RNS1^S306A^-FLAG*, respectively. Unless otherwise indicated, all fusion proteins are driven by the constitutive promoter *Ptef* from *Aureobasidium pullulans* in this study [28].

The binary plasmid pPK2-Sur-PRns1-GFP [28] was transformed into the mutant *∆**Slt2* to produce the strain *∆**Slt2-PRns1-gfp* (*gfp* driven by *PRns1* in the mutant *∆**Slt2*). The serine residue at position 306 in the genomic clone of *Rns1* (*gRns1*) [28] was mutated into alanine to produce the clone *gRns1^S306A^* with the Site-mutagenesis kit (NEB, Ipswich, MA, USA), which was then inserted into the plasmid pPK2-Sur-GFP [23] to form pPK2-Sur-GFP-gRns1^S306A^. The plasmid pPK2-Sur-GFP-gRns1^S306A^ was then transformed into the mutant *∆**Rns1* to produce the strain *∆**Rns1-gRns1^S306A^* (complementation of *∆**Rns1* using *gRns1* with the Ser-306 substituted to Ala). The primers used for production of the clone *gRns1^S306A^* were presented in Appendix A.

### 2.7. LC–MS/MS Analysis

To prepare the mycelium for LC–MS/MS analysis, 10^8^ conidia were inoculated into 100 mL of SDY medium and incubated at 26 °C for 36 h with 200 rpm shaking. The mycelium was collected by filtration and washed three times using sterile water. The sample was ground into fine powder using liquid nitrogen and suspended into the protein extraction buffer supplemented with protein inhibitors and phosphatase inhibitors (Sigma–Aldrich, St. Louis, MO, USA). The protein extraction buffer was prepared as previously described [28]. After 10 h incubation at 4 °C, the proteins were subjected to Western blot and SDS-PAGE analysis. The band corresponding to the fusion protein RNS1::FLAG was cut off from the SDS-PAGE gel and subjected to the LC–MS/MS analysis that was conducted by the company PTM-Bio (Hangzhou, China).

### 2.8. EMSA, ChIP-qPCR and qRT-PCR Analysis

EMSA (electrophoretic mobility shift assay assay) was conducted as previously described [26]. The expression and preparation of the protein RNS1-DBD (the truncated RNS1 protein contain DNA binding domain, Glu-61 to Pro-267) in the *E. coli* strain BL21 (DE3) were also conducted as previously described [28]. The biotin-labeled and unlabeled DNA probes were commercially synthesized (TsingKe Biological Technology, Hangzhou, China). The sequences of the DNA probes were presented in Appendix A. The EMSA assays were conducted using Light Shift Chemiluminescent EMSA kit (Thermo Fisher, Waltham, MA, USA).

ChIP-qPCR analysis was also conducted as previously described [28]. The enriched-DNA was analyzed by quantitative PCR analysis using the Thunderbird SYBR qPCR Mix without ROX (Toyobo, Japan). All EMSA and ChIP-qPCR assays were repeated three times.

For qRT-PCR analysis, Total RNA was extracted with Trizol reagent (Life Technologies, Carlsbad, CA, USA). The genes *act* and *tef* were used as references [22]. The relative normalized transcript level of a gene was computed using the 2^−∆∆Ct^ method [29]. cDNA synthesis was conducted using the ReverTra Ace qPCR RT Master Mix with a gDNA remover (Toyobo, Japan). qPCR was performed with the Thunderbird SYBR qPCR Mix (no ROX) (Toyobo, Japan). These experiments were repeated three times with three replicates per repeat.

### 2.9. RACE

The 5’RACE and 3’RACE (Rapid Amplification of cDNA Ends) assays were conducted using the RLM RT-PCR kit (Roche, Indiainapolis, IN, USA) that only clones intact mRNA with a 7-methyl guanosine cap structure. The primers used for RACE were shown in Appendix A.

## 3. Results

### 3.1. RNS1 Regulates Conidiation in M. robertsii

In our previous study, we found that the transcription factor RNS1 regulates the utilization of less-favored carbon and nitrogen sources by *M. robertsii* [28]. In this study, we found that RNS1 controlled conidiation on the PDA plates with the favored carbon source (glucose). No significant difference in colony growth rate was found between the wild-type strain (WT) and the deletion mutant of *Rns1* (*∆Rns1*), but the mutant *∆**Rns1* produced a “fluffy” colony phenotype with reduced conidial yield (Figure 1A,B). For the WT strain, aerial hyphae were highly branched and differentiated to form clustered phialides where the chained conidia were produced. The development of conidiophores of the mutant *∆**Rns1* was impaired, with fewer phialides formed (Figure 1C). For all above assays, no significant difference was found between the WT strain and the complemented strain *C-∆**Rns1*. The genes and fungal strains used in this study are listed in Table 1.

The relative germination inhibition was used to compare the conidial germination of the WT strain, the deletion mutant *∆**Rns1* and the complemented strain *C-∆**Rns1* under several abiotic stresses. No difference in tolerance to UV radiation, hyperosmotic stress, oxidative stress and cell wall-disturbing agent was found between the three strains (Appendix A).

qRT-PCR analysis was conducted to assay the expression of *Rns1* during the vegetative growth (at two days post inoculation (DPI) by evenly spreading 100 μL of a conidial suspension (1 × 10^7^ conidia/mL) on a PDA plate (90 mm)), the early conidiation stage (at three DPI), the middle conidiation stage (at five DPI), and the late conidiation stage (at seven DPI). The conidiation stages were determined as previously described [21,30]. Compared to the vegetative stage, qRT-PCR analysis showed that the expression level of *Rns1* was increased at three and five DPI and reduced at the late stage of conidiation (Figure 1D).

### 3.2. RNS1 Positively Regulates the Central Regulatory Pathway for Conidiation

As in many other fungi, the central regulatory pathway containing BrlA, AbaA and WetA is also essential for conidiation in *M. robertsii* [21]. We identified a putative RNS1 binding motif *BM2* in the promoter regions of the genes *BrlA* (ACCAGAC) (MAA_10599) and *AbaA* (ACAAGAC) (MAA_00694). EMSA was conducted to analyze the binding of the recombinant protein RNS1-DBD to the probes containing the putative *BM2* motif in *BrlA* and *AbaA* promoters. For the biotin-labeled probe from the *BlrA* promoter, the protein RNS1-DBD shifted its migration, and the specific competitor (unlabeled DNA probe) abolished such band shift (Figure 2A). However, the RNS1-DBD protein had no impact on the migration of the biotin-labeled probe from the *AbaA* promoter (Figure 2B).

An ChIP-qPCR assay showed that the copy number of the enriched DNA fragment containing the *BM2* motif in the *BrlA* promoter in the strain *WT-RNS1-FLAG* (expresses the fusion protein RNS1::FLAG with RNS1 tagged with FLAG) was 103-fold higher than *WT-FLAG* strain that expressed the FLAG tag only (Figure 2C). But the ChIP-qPCR assay showed that the RNS1::FLAG protein did not bind to the *AbaA* promoter in vivo (Figure 2D).

### 3.3. RNS1 Induces the Expression of the BrlAα Transcript

We then investigated how RNS1 regulated the expression of the *BrlA* gene. As described above, the *BrlA* gene has two overlapping transcripts (*BrlAα* and *BrlAβ*) in the fungus *A. nidulans* [8,31]. We thus assayed whether the *BrlA* gene also had two transcripts in *M. robertsii* using the 5′ and 3′ RACE. Only one PCR band was cloned with the 3′RACE, while two bands were obtained with the 5′RACE (Figure 3A). All PCR products were cloned, sequenced and assembled, and we found that the *BrlA* gene in *M. robertsii* also had two overlapping transcripts, with the long one named as *BrlAβ* (Genbank accession number: OL739242) and the short one as *BrlAα* (Genbank accession number: OL739243). The transcripts *BrlAα* and *BrlAβ* contain an identical ORF (designated as major ORF) that encodes BrlA (Figure 3B), and they differed only in that the 5′UTR of *BrlAβ* was 835 bp longer than that of *BrlAα*. The transcription start sites of the transcripts *BrlAα* and *BrlAβ* were at −1086 and −251 bp upstream of the start codon of the major ORF, respectively (Figure 3B). The RNS1 binding motif BM2 was at −1104 bp upstream of the major ORF. In the 5′UTR of the transcript *BrlAβ*, multiple upstream ORFs (uORFs) were predicted, but none of the uORFs encoded proteins that had homologs in the non-redundant protein database (nr) at NCBI.

Using two sets of primers, qRT-PCR analysis was conducted to compare the expression levels of the transcripts *BrlAα* and *BrlAβ* between the WT strain and the deletion mutant *∆Rns1*. With the Primer set 1 for the detection of the expression level of *BrlAβ*, no significant difference was found between the WT strain and the mutant *∆Rns1*. However, when the Primer set 2 was used to assay both *BrlAα* and *BrlAβ*, the expression of the two transcripts in the WT strain was 9.1-fold higher than the mutant *∆Rns1*. Taken together, RNS1 induced the expression of the transcript *BrlAα* (Figure 3C).

As previously described, in the central regulatory pathway, BrlA activates the expression of *AbaA*, which in turn induces *WetA* expression [21]. We thus further investigated whether the regulation of *BrlA* by RNS1 impacted the expression of *AbaA* and *WetA*. qRT-PCR analysis showed that *AbaA* was 10-fold more highly expressed in the WT strain than the mutant *∆Rns1*, and the *WetA* expression was also reduced in the mutant *∆Rns1*. (Figure 3D).

### 3.4. Slt2-MAPK Phosphorylates RNS1 during Conidiation

To identify the upstream signaling components that regulate *Rns1* during conidiation, the expression of *Rns1* was assayed in the mutants of the previously characterized signaling components that are involved in conidiation [23]. In our previous study, we found that the Fus3-MAPK regulates pigmentation during conidiation and activates *Rns1* for utilizing less-favored carbon and nitrogen sources [22,23,28]. In this study, we found that Fus3-MAPK did not regulate *Rns1* expression during conidiation on PDA plates (Figure 4). The deletion mutants of two other MAPKs (Slt2 and Hog1) encoding genes were also impaired in conidiation [23]. qRT-PCR analysis showed that Hog1-MAPK had no impact on *Rns1* expression during conidiation (Figure 4), but *Rns1* was over 3-fold more highly expressed in the WT strain than the deletion mutant *∆**Slt2* and the mutants of MAPKK (*∆**Mkk*) and MAPKKK (*∆**Bck1*) in the Slt2-MAPK cascade (Figure 4), indicating that the Slt2-MAPK cascade activates *Rns1* expression during conidiation. The deletion mutants *∆**Slt2*, *∆**Mkk*, *∆**Bck1* and *∆**Hog1* were constructed in our previously study [23]. qRT-PCR analysis further showed that the genes *BrlA* and *AbaA* were more highly expressed in the WT strain than the mutant *∆**Slt2*, but the expression level of *WetA* in *∆**Slt2* was higher than the WT strain (Figure 3D).

We then assayed whether the kinase Slt2 phosphorylated the RNS1 protein. An LC–MS/MS analysis showed that the serine residue at position 306 in the fusion protein RNS1::FLAG was phosphorylated in the *WT-RNS1-FLAG* strain (Figure 5A).

We then investigated whether Slt2 directly phosphorylated the serine residue at position 306. A yeast two-hybrid assay showed that Slt2 directly interacted with RNS1 (Figure 5B). Using the strain *RNS1-FLAG/Slt2-Myc* that expressed the fusion protein RNS1::FLAG and Slt2::Myc (a protein with Slt2 tagged with Myc), a Co-IP assay confirmed that RNS1 physically contacted Slt2 in vivo (Figure 5C).

Phos-tag assay was further conducted to confirm the phosphorylation by Slt2 of the serine residue at position 306 in the RNS1 protein. During conidiation, the Phos-tag assay showed that there was only one protein band in the strains *∆**Slt2-RNS1-FLAG*, *WT-RNS1^S306A^-FLAG* and *∆Slt2-RNS1^S306A^-FLAG*, while two different bands were obviously seen in the strain *WT-RNS1-FLAG* (Figure 5D), which corresponded to the protein RNS1::FLAG with different extent of phosphorylation of the serine residue at position 306.

Western blotting analysis was conducted to assay the impact of phosphorylation of the Ser-306 by Slt2 on the stability of the RNS1 protein. No significant difference in the expression level of the RNS1::FLAG protein was found between the strains *WT-RNS1-FLAG*, *WT-RNS1^S306A^-FLAG*, *∆Slt2-RNS1-FLAG*, *∆Slt2- RNS1^S306A^-FLAG* (Figure 5E).

### 3.5. Phosphorylated RNS1 Upregulates Its Own Expression during Conidiation

In our previous study [28], we found that *Rns1* self-induces expression during infection of insects. We thus investigated whether it also upregulated its own expression during conidiation. To this end, we first used ChIP-qPCR to investigate whether phosphorylation of RNS1 by Stl2 impacted the binding of RNS1 to the *BM2* motif in the *Rns1* promoter. Compared to the strain *WT-FLAG*, nearly three-fold more DNA fragment containing the *BM2* motif in the *Rns1* promoter was enriched in the strain *WT-RNS1-FLAG*; no significant difference was found between the strains *WT-FLAG*, *WT-RNS1^S306A^-FLAG* and *∆**Slt2-RNS1-FLAG* (Figure 6A).

Another assay was conducted to confirm that *Rns1* self-induces expression. In addition to the previously published strains *WT-PRns1-gfp* (the *gfp* gene driven by the *Rns1* promoter *PRns1* in the WT strain), *WT-PRns1^∆BM2^-gfp* (*gfp* driven by a mutated *PRns1* with the *BM2* motif mutated in the WT) and *∆**Rns1-PRns1-gfp* (*gfp* driven by *PRns1* in the mutant *∆Rns1*) [28], we constructed another strain *∆**Slt2-PRns1-gfp* (*gfp* driven by *PRns1* in the mutant *∆**Slt2*). During conidiation, the GFP signal intensity was the strongest in the strain *WT- PRns1-gfp*, and no obvious GFP signal was seen in the strains *WT-PRns1^∆BM2^-gfp*, *∆**Rns1-PRns1-gfp* and *∆**Slt2-PRns1-gfp* (Figure 6B). Consistent with the intensity of the GFP signal, the expression levels of GFP transcript and protein in the strain *WT-PRns1-gfp* were also higher than the strains *WT-PRns1^∆BM2^-gfp*, *∆**Rns1-PRns1-gfp* and *∆**Slt2-PRns1-gfp* (Figure 6C,D).

We further investigated the roles in *Rns1* self-induction of the Ser-306 and the *BM2* motif in the *Rns1* promoter. To this end, in addition to the two previously published strains C-*∆Rns1* (the complemented strain of *∆Rns1*) and *∆**Rns1-gRns1^∆BM2^* (complementation of *∆**Rns1* using the *Rns1* genomic clone *gRns1* with motif *BM2* mutated) [28], *∆**Rns1-gRns1^S306A^* (complementation of *∆**Rns1* using *gRns1* with the Ser-306 substituted to Ala) was also constructed. Compared to the WT strain, the expression level of *Rns1* was reduced 100-fold and 50-fold in the strain *∆**Rns1-gRns1^∆BM2^* and *∆**Rns1-gRns1^S306A^*, respectively; no difference was found between the WT strain and the complemented strain C-*∆Rns1*. (Figure 6E).

## 4. Discussion

Several upstream signaling regulators have been documented to control conidiation in Ascomycete fungi, which is essential for fungal long-term survival under a variety of environmental conditions [6,13]. In this study, we found that the Slt2-MAPK signaling pathway and the transcription factor RNS1 constitute a novel upstream signaling cascade that activates the central regulatory pathway for conidiation in the Ascomycetes fungus *M. robertsii*. Slt2 phosphorylates the serine residue at position 306 in the transcription factor RNS1, which self-induces and in turn directly induces the expression of BlrAα, the initiator of the central regulatory pathway. The roles of the Slt2-MAPK signaling pathway in conidiation diversified in this phylum. Slt2-MAPK plays important roles in conidiation in many fungi such as *Botrytis cinerea*, *A. oligospora*, *M. acridum* and *Monacrosporium haptotylum* [32,33], but it is indispensable for conidiation in other fungi such as *Aspergillus fumigatus* [34]. Therefore, the regulation of conidiation by the Slt2/RNS1 cascade could only function in a subgroup of ascomycetes fungi. To some extent, the central regulatory pathway for conidiation diversified among the Ascomycete fungi. In *Aspergillus* fungi, the transcript *BrlAβ* encodes a protein that is 23-aa longer than that of *BrlA**α* [8], while *BrlAα* and *BrlAβ* in *M. robertsii* have the same ORF.

In addition to conidiation, deletion of the gene *Slt2* also reduced the tolerance to hyperosmotic stress and increased sensitivity to the cell wall-disturbing agent Congo red [23]. However, no difference in tolerance to hyperosmotic stress and cell wall-disturbing agent was found between the WT strain and the deletion mutant *∆Rns1.* Therefore, RNS1 is one of the transcription factors regulated by the Slt2-MAPK signaling pathway.

RNS1 also showed at least two regulatory roles in *M. robertsii*, one is, as described above, the regulation of conidiation on the medium containing a favored carbon source (glucose). The other one is to regulate the utilization of less-favored carbon and nitrogen sources, which is controlled by the Fus3-MAPK signaling [28]. As a direct target of two MAPK signaling pathways, the transcription factor RNS1 appears to be a central regulator that controls fungal responses to multiple environmental clues. In response to less-favored carbon and nitrogen sources, the Fus3-MAPK phosphorylates the serine at position 226 and threonine at position 215 in RNS1 to activate genes for the utilization of these nutrients [28]. During conidiation, the serine at position 306 is phosphorylated by the Slt2-MAPK, which results in the activation of the central regulatory pathway for conidiation. The deletion mutant of *BrlA* completely lost the ability to produce conidiophores, while the deletion mutant of *Rns1* was still able to produce conidia though it had a significantly lower conidial yield than the WT strain, which could be attributed to the fact that RNS1 only regulates the expression of the transcript *BrlAα*. The transcript *BrlAβ* is supposed to have a function during conidiation, and the relationship between the transcripts *BrlAα* and *BrlAβ* remains to be clarified.

In conclusion, in the Ascomycete fungus *M. robertsii*, the Slt2/RNS1 cascade represents a novel upstream regulator that initiates conidiation via direct activation of the central regulatory pathway. This work provides significant mechanistic insights into the production of asexual conidia in the Ascomycete fungi.

## Figures and Tables

**Figure 1 jof-08-00026-f001:**
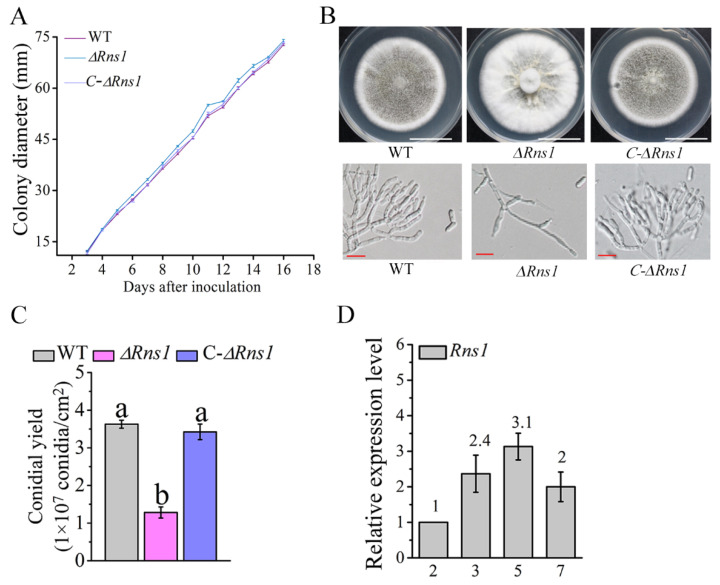
RNS1 regulates conidiation. (**A**) Colony growth curve on PDA plates. WT: the wild-type strain; *∆**Rns1*: the deletion mutant of *Rns1*; *C-∆**Rns1*: the complemented strain of *∆**Rns1*. Data are shown as the mean ± SE. (**B**) Colony morphology (upper panel; scale bar: 3 cm) and conidiophores (lower panel; scale bar: 10 μm) on the PDA plates. In this study, all shown images are representative of at least three independent experiments. (**C**) Conidial yields. Conidial yield assay was repeated three times with three replicates. Data are shown as the mean ± standard error (SE). Values with different letters are significantly different (*p* < 0.05, Tukey’s test in one-way ANOVA). (**D**) qRT-PCR analysis of the expression of *Rns1* at the four conidiation stages of the WT strain on PDA plates. The values represent the fold-change of expression of *Rns1* at a conidiation stage compared with the expression at 2 days post-inoculation, which is set to 1.

**Figure 2 jof-08-00026-f002:**
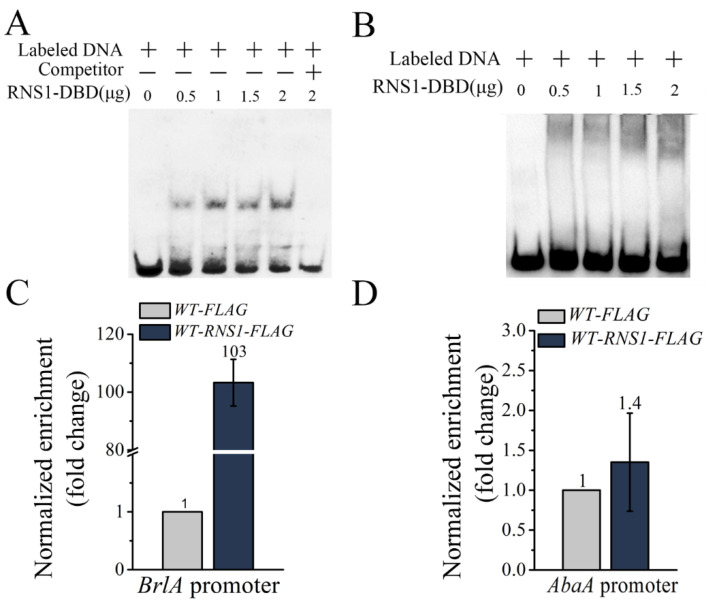
EMSA and ChIP-qPCR analysis of the binding of RNS1 to the putative the *BM2* motifs in the promoters of the gene *BrlA* and *AbaA*. (**A**) EMSA analysis of the binding of the recombinant protein RNS1-DBD (the DNA binding domain in RNS1) to the Biotin-labeled *BM2* motif in the *BrlA* promoter, and (**B**) in the *AbaA* promoter. The binding activity was demonstrated by the shift of the labeled DNA band prior to the addition of the specific competitor (the unlabeled DNA probe) in a 200-fold excess. (**C**) ChIP-qPCR analysis of the binding of the fusion proteins RNS1::FLAG to the *BrlA* promoter, and (**D**) to the *AbaA* promoter. *WT-FLAG:* a strain expressing the tag FLAG in the WT strain; *WT-RNS1-*FLAG: a strain expressing the fusion protein RNS1::FLAG protein in the WT.

**Figure 3 jof-08-00026-f003:**
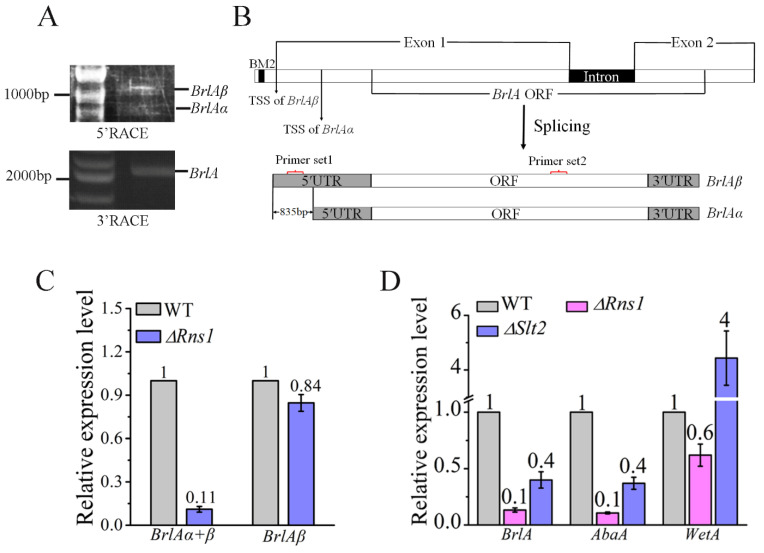
Identification and characterization of the transcripts *BrlAα* and *BrlAβ* of the *BrlA* gene. (**A**) 5′RACE (upper panel) and 3′RACE (lower panel) amplification of the *BrlA* transcripts using the RLM RT-PCR kit. Note: two PCR products were obtained with 5′RACE. (**B**) A diagrammatic representation of the genome sequence of the gene *BrlA* (upper panel) and its two transcripts (*BrlAα* and *BrlAβ*) (down panel). The position of the *BM2* motif is shown. Note: the two transcripts have the same major ORF and different 5′UTR. TSS: transcription start site. (**C**) qRT-PCR analysis of the expression of *BrlAβ* with Primer set 1, and of *BrlAα* and *BrlAβ* using the Primer set 2 during conidiation. The relative positions of the primers are shown in (B). (**D**) qRT-PCR analysis of the expression level of *BrlA* (the Primer set 2), *AbaA* and *WetA* in WT, the deletion mutants *∆Rns1* and *∆Slt2*. The values in each figure represent the fold-changes of expression of a gene in a mutant compared with WT, which is set to 1.

**Figure 4 jof-08-00026-f004:**
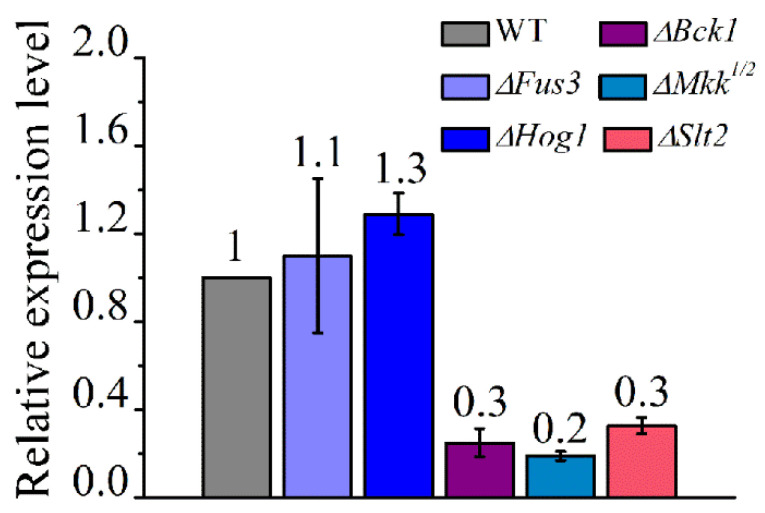
qRT-PCR analysis of the expression of the gene *Rns1* during conidiation in the WT strain, the deletion mutant *∆Fus3* (Fus3-MAPK), *∆Hog1* (Hog1-MAPK) and the three deletion mutants in the Slt2-MAPK cascade: *∆**Bck1* (MAPKKK), *∆**Mkk^1/2^* (MAPKK) and *∆**Slt2* (MAPK). The values represent the fold-changes of *Rns1* expression in a mutant compared with WT, which is set to 1.

**Figure 5 jof-08-00026-f005:**
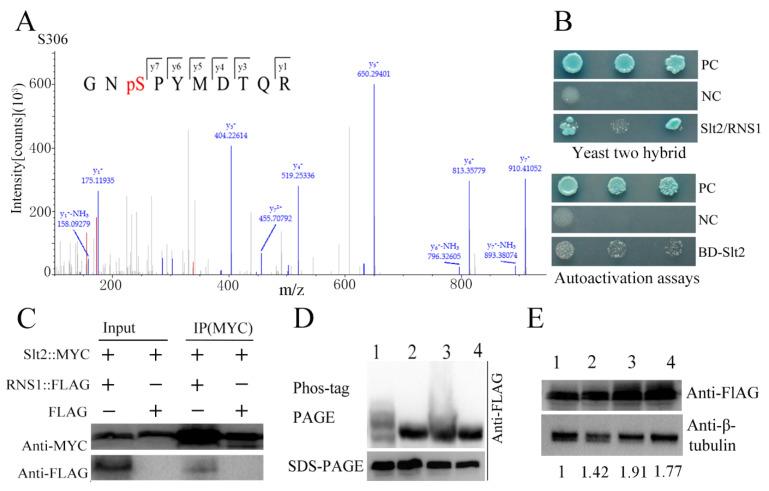
Phosphorylation of RNS1 by Slt2 during conidiation. (**A**) Phosphorylated peptide spectra identified by LC–MS/MS. The serine residue at position 306 was phosphorylated. The *WT-RNS1-FLAG* strain was used to conduct mass spectrometry assay. (**B**) Yeast two-hybrid analysis showing the physical interaction between the Slt2 and RNS1. Upper panel: colonies grown in SD-His-Ade-Leu-Trp + X-α-gal + AbA. Slt2/RNS1: cells expressing Slt2 and RNS1. Lower panel: colonies grown in SD-His-Trp-Ade plus X-a-Gal. BD-Slt2:: Y2HGold cells expressing Slt2; NC: negative control; PC: positive control. (**C**) Co-IP confirmation of the interaction of Slt2 and RNS1. Immunoprecipitation was conducted with anti-Myc antibody. Proteins were detected by immunoblot analysis with anti-Myc and anti-FLAG antibodies. (**D**) Phos-tag analysis of RNS1 phosphorylation by Slt2. Proteins were detected with the anti-FLAG antibody. (1) *WT-RNS1-FLAG* (RNS1::FLAG expressed in WT strain); (2) *WT-RNS1^S306A^-FLAG* (RNS1^S306A^::FLAG (Ser-306 substituted to Ala in RNS1::FLAG) expressed in the WT strain); (3) *∆Slt2-RNS1-FLAG* (RNS1::FLAG expressed in *∆**Slt2*); (4) *∆Slt2-**RNS1^S306A^-FLAG* (RNS1^S306A^::FLAG expressed in *∆**Slt2*. (**E**) Western blot analysis of the protein RNS1::FLAG in the strains *WT-RNS1-FLAG*, *WT-RNS1^S306A^-FLAG*, *∆Slt2-RNS1-FLAG* and *∆**Slt2-RNS1^S306A^-FLAG*.

**Figure 6 jof-08-00026-f006:**
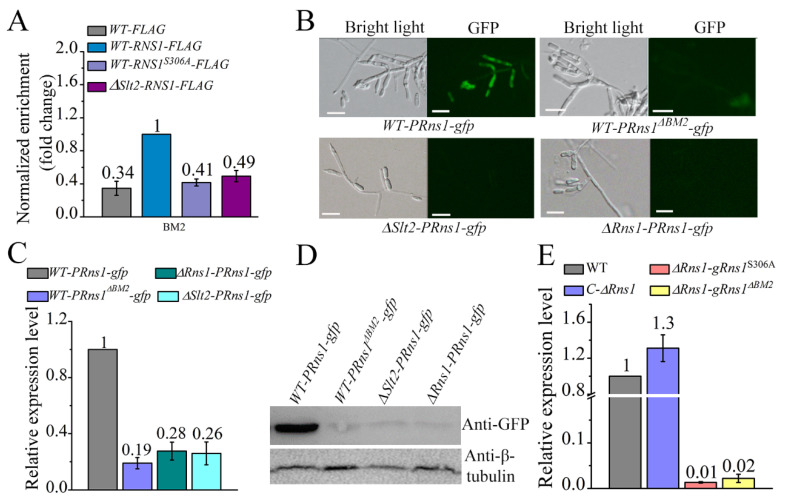
Phosphorylation of the Sec-306 of RNS1 by Slt2 facilitates the binding of RNS1 to its own promoter to self-induce expression during conidiation. (**A**) ChIP-qPCR analysis of the binding to the *BM2* motif in the *Rns1* promoter of the proteins RNS1::FLAG or RNS1^S306A^::FLAG. *WT-FLAG:* a strain expressing the tag FLAG in the WT strain; *WT-RNS1-*FLAG: a strain expressing the fusion protein RNS1::FLAG protein in the WT. *WT-RNS1^S306A^-FLAG*: a strain expressing RNS1^S306A^::FLAG protein in WT. *∆Slt2-RNS1-FLAG*: a strain expressing RNS1::FLAG protein in *∆**Slt2.* Microscopic observation (**B**), qRT-PCR analysis (**C**) and Western blot analysis (**D**) of GFP expression during conidiation. *WT-PRns1-gfp*: a strain with *gfp* driven by the native *Rns1* promoter (*PRns1*) in WT; *WT-PRns1^BM2^-gfp*: *gfp* driven by *PRns1^∆BM2^* (the *BM2* motif was mutated in *Rns1*’s promoter *PRns1*) in WT; *∆**Rns1-PRns1-gfp: gfp* driven by the promoter *PRns1* in *∆**Rns1*; *∆**Slt2-PRns1-gfp*: *gfp* driven by the promoter *PRns1* in *∆**Slt2*. Bars, 10 μm. (**E**) qRT-PCR analysis of the expression of *Rns1**. ∆**Rns1-gRns1^S306A^*: a strain constructed by complementation of the mutant *∆**Rns1* with a mutated genomic clone of *Rns1* (*gRns1*) with the serine residue at position 306 changed to alanine; *∆**Rns1-gRns1^∆BM2^*: a strain constructed by complementation of the mutant *∆**Rns1* with a *gRns1* mutant with the *BM2* motif mutated.

**Table 1 jof-08-00026-t001:** Plasmids, fusion proteins and fungal strains used in this study.

Name	Description	Ref
**Fusion proteins**		
RNS1::FLAG	RNS1 tagged with FLAG	[28]
RNS1-DBD	A section of RNS1 containing DNA binding domain	[28]
RNS1^S306A^::FLAG	A mutated RNS1::FLAG with Ser-306 substituted to Ala	This study
Slt2::MYC	Slt2 tagged with MYC	This study
Plasmids		
pPK2-bar-Ptef-MYC	Expression of a protein tagged with MYC	[28]
pPK2-sur-Ptef-FLAG	Expression of a protein tagged with FLAG	[28]
pPK2-sur-Ptef-RNS1-FLAG	Expression of the protein RNS1::FLAG	This study
pPK2-bar-Ptef-Slt2-MYC	Expression of the protein Slt2::MYC	This study
pPK2-sur-Ptef-RNS1^S306A^-FLAG	Expression of the protein RNS1^S306A^::FLAG	This study
**Promoters**		
*PRns1*	The promoter region (1724 bp upstream of the ORF) of the gene *Rns1*	[28]
*PRns1^∆BM2^*	The mutated *PRns1* with all 7 nt in the *BM2* motif substituted to A	[28]
*PbrlA* *PAbaA*	The promoter region (2000 bp upstream of the ORF) of the gene *BrlA*The promoter region (1020 bp upstream of the ORF) of the gene *AbaA*	This study This study
**Genomic clones**		
*gRns1*	The genomic clone of the *Rns1* gene	[28]
*gRns1^S306A^*	A mutated *gRns1* with Ser-306 substituted to Ala in the RNS1 protein	This study
*gRns1* * ^∆^ * * ^BM2^ *	A mutated *gRns1* with all 7 nt in the *BM2* motif in *PRns1* substituted to A	[28]
**Fungal strains**		
WT	The wild-type strain of *M. robertsii* ARSEF2575	[28]
*∆* *Rns1*	The deletion mutant of the *Rns1* gene	[28]
*C-* *∆* *Rns1*	The complemented strain of the mutant *∆**Rns1*	[28]
*WT-FLAG*	Expressing FLAG tag in the WT strain	[28]
*WT-RNS1-FLAG*	Expressing RNS1::FLAG in the WT strain	[28]
*∆* *Slt2-* *RNS1-FLAG*	Expressing RNS1::FLAG in the mutant *∆**Slt2*	This study
*WT-RNS1^S306A^-FLAG*	Expressing RNS1^S306A^::FLAG in the WT strain	This study
*∆* *Slt2-* *RNS1^S306A^-FLAG*	Expressing RNS1^S306A^::FLAG in the mutant *∆**Slt2*	This study
*WT-PRns1-gfp*	The *gfp* gene driven by *PRns1* in the WT strain	[28]
*WT-PRns1* * ^∆^ * * ^BM2^ * *-gfp*	The *gfp* gene driven by *PRns1**^∆^**^BM2^* in the WT strain	[28]
*∆* *Rns1-PRns1-gfp*	The *gfp* gene driven by *PRns1* in the mutant *∆**Rns1*	[28]
*∆* *Slt2-PRns1-gfp*	The *gfp* gene driven by *PRns1* in the mutant *∆**Slt2*	This study
*RNS1-FLAG/Slt2-MYC*	Expressing RNS1::FLAG and Slt2::MYC in the WT strain	This study
*RNS1-FLAG/MYC*	Expressing RNS1::FLAG and MYC tag in the WT strain	This study
*∆* *Rns1-gRns1^S306A^*	Complementation of *∆**Rns1* using the genomic clone *gRns1 ^S306A^*	This study
*∆* *Rns1-gRns1* * ^∆^ * * ^BM2^ *	Complementation of *∆**Rns1* using the genomic clone *gRns1**^∆^**^BM2^*	This study

## Data Availability

Not applicable.

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
