# Peer review of "Slt2-MAPK/RNS1 Controls Conidiation via Direct Regulation of the Central Regulatory Pathway in the Fungus *Metarhizium robertsii"

_jof, 2021, doi:10.3390/jof8010026_

Round 1

Reviewer 1 Report

In the manuscript entitled “Slt2-MAPK/RNS1 controls conidiation via direct regulation of the central regulatory pathway in the fungus Metarhizium robertsii” Meng et al. report the functional characterization of the BrlA transcription factor. They demonstrate that the corresponding BrlA locus encodes two overlapping transcripts. One of them encodes for a protein that is phosphorylated by the Slt2-MAPK. The phosphorylated serine residue is located into the RNS1 motif which binds to DNA. The binding of the BrlA transcription factor results in activation of its own expression as well as those of the AbaA and WetA genes, which are required for conidiation.

This work is of interest to a wide audience of molecular microbiologists/mycologists and fully corresponds to the scope of Journal of Fungi. This well-conducted, concise and focused study provides new insights into the mechanisms responsible for conidiation. The experiments are sound, robust for the most part and support the conclusions reached by the authors. However, to be suitable for publication, this work should be revised considering the following remarks:

Major points :

- The term "central regulatory pathway" used throughout the manuscript, is ill-defined and does not correspond to a clearly identified entity. This expression should be rephrased in a more precise manner.

- Figure 1A : There is no standard deviation (+/- SD) on the colony growth curve. Have biological replicates been performed? If so, why is the standard deviation not indicated?

- Figure 2A : Below the main band described in the text, it appears that a second band is visible in wells 2-5. If this is the case, it would mean that the main band would be the product of a complex binding (super-shift) and that the faint band at the bottom would correspond to a monomer binding (shift). This point needs to be further clarified.

- Figure 3A : the picture of the 5’RACE experiment is of poor quality. This should be improved.

- Figure 3B: the schematic representation of the BrlA locus is mis-annotated. Even if two transcriptional start sites (TSS) are present, there is only one 5’UTR sequence for the BrlA gene . Hence, these TSS should be clearly represented on the diagram since they are missing one the current version. Moreover the “+1” indication on the lower panel is also misleading because it refers to the ATG whereas by convention, it should refer to the +1 of transcription (TSS). The diagram should be edited accordingly.

- Figure 5D: lane 1, the pattern is fuzzy. Does this correspond to several discrete bands as indicated in the text or just a smear? This point needs to be further clarified.

Minor points :

- p1 line 39: reference needed.

- p2 line 56: the term “several upstream components” is too vague. This statement needs to be rephrase in a more precise manner.

- p2 line 61: space required

- p2 line 65-66 : In the introduction, a few words about phylogenetic position of M. robertsii compared to Aspergilus species would be welcomed.

- p4 line 183: space required

- p6 line 196 : “relative germination inhibitor” is confusing. Why not refer to the classic germination rate?

- p6 line 204 : What is the meaning of “mature” in this context?

- Figure 2B : what is the meaning of “a” and “b” located on the tops of histograms?

- Figure 2 Legend (line 237) : DNA instead of DNB?

Author Response

Reviewer 1:

Comments and Suggestions for Authors

Comment 1: The term "central regulatory pathway" used throughout the manuscript, is ill-defined and does not correspond to a clearly identified entity. This expression should be rephrased in a more precise manner.

Response: Rephrased for clarity.

Comment 2:  Figure 1A: There is no standard deviation (+/- SD) on the colony growth curve. Have biological replicates been performed? If so, why is the standard deviation not indicated?

Response: This experiment was repeated three times. Standard errors were shown in the Figure 1A in the first version of this figure, which were two small to see. In the revised figure, we changed the width of the lines so that the standard errors can be seen.

Comment 3: Figure 2A: Below the main band described in the text, it appears that a second band is visible in wells 2-5. If this is the case, it would mean that the main band would be the product of a complex binding (super-shift) and that the faint band at the bottom would correspond to a monomer binding (shift). This point needs to be further clarified.

Response: It seems to us that “the second band”was looming, and we are not sure it is a real band. In addition, this possible band was not always seen in other repeats of this experiment, so we decided not to state that there was a monomer binding.

Comment 4: Figure 3A: the picture of the 5’RACE experiment is of poor quality. This should be improved.

Response: Sometimes, it is hard to get strong bands when the 5¢UTRs of the intact mRNAs are cloned with the RLM-RT-PCR kit. We have tried twice to do this experiment, but got the same result. So, for the BrlA gene in M. robertsii, it may be not possible to get a picture with bright bands. However, the two bands were confirmed to be two 5¢UTRs of the BrlA gene by sequencing.

Comment 5: Figure 3B: the schematic representation of the BrlA locus is mis-annotated. Even if two transcriptional start sites (TSS) are present, there is only one 5’UTR sequence for the BrlA gene . Hence, these TSS should be clearly represented on the diagram since they are missing one the current version. Moreover the “+1” indication on the lower panel is also misleading because it refers to the ATG whereas by convention, it should refer to the +1 of transcription (TSS). The diagram should be edited accordingly.

Response: modified for clarity.

Comment 6: Figure 5D: lane 1, the pattern is fuzzy. Does this correspond to several discrete bands as indicated in the text or just a smear? This point needs to be further clarified.

Response: This point was further clarified in the revised manuscript. Multiple bands in the Lane 1 were really discrete bands, which was not a smear. The same result was obtained in three independent repeats of this experiment, which is also typic in many previously documented Phos-tag experiments such as the reference 26 in this study.

Comment 7: p1 line 39: reference needed.

Response: reference added.

Comment 8: p2 line 56: the term “several upstream components” is too vague. This statement needs to be rephrase in a more precise manner.

Response: Rephrased

Comment 9: p2 line 61: space required

Response: Added

Comment 10: p2 line 65-66 : In the introduction, a few words about phylogenetic position of M. robertsii compared to Aspergilus species would be welcomed.

Response: added

Comment 11: p4 line 183: space required

Response: added

Comment 12: p6 line 196 : “relative germination inhibitor” is confusing. Why not refer to the classic germination rate?

Response: The germination rates of the WT strain and the deletion mutant DRns1 were not the same under optimal conditions. To compare the impact of the stresses on the germination of these two strains, relative germination inhibition by the stress was used to normalize the difference in germination between the strains. Relative germination inhibition was also used in other studies.

Comment 13: p6 line 204 : What is the meaning of “mature” in this context?

Response: “Mature”is the late conidiation stage when no more conidia are produced on the PDA plates. We revised this sentence for clarity.

Comment 14: Figure 2B : what is the meaning of “a” and “b” located on the tops of histograms?

Response: We assumed the reviewer was talking about “a” and “b” located on the tops of histograms in Figure 1c since these were no such labels in Figure 2B. They showed the significant difference in conidial yield between the WT and the mutant DRns1, which is defined in the figure legend.

Comment 15: Figure 2 Legend (line 237) : DNA instead of DNB?

Response: changed.

Reviewer 2:

Comments and Suggestions for Authors

Comment 1: The authors needs to explain their aims in more detail. 

Response: More information is added to clarity our aims.

Comment 2: The authors need to check for errors, for example RNS1regulates (Line 183).

Response: thoroughly corrected.

Comment 3: The authors need to report their results. For example, they start paragraphs with "As previously described (Line 201) and "we then tried to identify" (Line 284).

Response: The text is revised for a better presentation.

Comment 4: The manuscript only consist of 3 paragraphs. If the authors can elaborate on the discussion in the revised, especially compare with previous studies.

Response: I believe that reviewer meant that the discussion only consists of three paragraphs. We have thought carefully about the suggestion about expansion of the discussion. The Slt2-MAPK/RNS1 cascade represents a novel regulatory mechanism of the central regulatory pathway during conidiation. We pointed out the novelty of this finding in the first paragraph. Slt2-MAPK and RNS1 were discussed along with previous studies in the second and third paragraph, respectively. We cannot find the relationship between the Slt2-MAPK/RNS1 cascade and the previously published signaling components, which cannot be properly discussed.  Therefore, for the time being, we cannot further elaborate on the discussion.

Reviewer 2 Report

The authors studied how Slt2-MAPK/RNS1 controls conidiation via direct regulation of 2 the central regulatory pathway in the fungus Metarhizium robertsii. The work is important and should be published.

However, the following changes needs to be made to be accepted for publication.

  1. The authors needs to explain their aims in more detail. 
  2. The authors need to check for errors, for example RNS1regulates (Line 183).
  3. The authors need to report their results. For example, they start paragraphs with "As previously described (Line 201) and "we then tried to identify" (Line 284).
  4. The manuscript only consist of 3 paragraphs. If the authors can elaborate on the discussion in the revised, especially compare with previous studies.

Author Response

Reviewer 2:

Comments and Suggestions for Authors

Comment 1: The authors needs to explain their aims in more detail. 

Response: More information is added to clarity our aims.

Comment 2: The authors need to check for errors, for example RNS1regulates (Line 183).

Response: thoroughly corrected.

Comment 3: The authors need to report their results. For example, they start paragraphs with "As previously described (Line 201) and "we then tried to identify" (Line 284).

Response: The text is revised for a better presentation.

Comment 4: The manuscript only consist of 3 paragraphs. If the authors can elaborate on the discussion in the revised, especially compare with previous studies.

Response: I believe that reviewer meant that the discussion only consists of three paragraphs. We have thought carefully about the suggestion about expansion of the discussion. The Slt2-MAPK/RNS1 cascade represents a novel regulatory mechanism of the central regulatory pathway during conidiation. We pointed out the novelty of this finding in the first paragraph. Slt2-MAPK and RNS1 were discussed along with previous studies in the second and third paragraph, respectively. We cannot find the relationship between the Slt2-MAPK/RNS1 cascade and the previously published signaling components, which cannot be properly discussed.  Therefore, for the time being, we cannot further elaborate on the discussion.